# Clinical and Genetic Insights into Desbuquois Dysplasia: Review of 111 Case Reports

**DOI:** 10.3390/ijms25179700

**Published:** 2024-09-07

**Authors:** Hubert Piwar, Michal Ordak, Magdalena Bujalska-Zadrozny

**Affiliations:** Department of Pharmacotherapy and Pharmaceutical Care, Faculty of Pharmacy, Medical University of Warsaw, Banacha 1 Str., 02-097 Warsaw, Poland; hubertpiwar@gmail.com (H.P.); magdalena.bujalska@wum.edu.pl (M.B.-Z.)

**Keywords:** Desbuquois dysplasia, *CANT1* gene

## Abstract

Skeletal disorders encompass a wide array of conditions, many of which are associated with short stature. Among these, Desbuquois dysplasia is a rare but severe condition characterized by profound dwarfism, distinct facial features, joint hypermobility with multiple dislocations, and unique vertebral and metaphyseal anomalies. Desbuquois dysplasia is inherited in an autosomal recessive manner, with both the DBQD1 (MIM 251450) and DBQD2 (MIM 615777) forms resulting from biallelic mutations. Specifically, DBQD1 is associated with homozygous or compound heterozygous mutations in the *CANT1* gene, while DBQD2 can result from mutations in either the *CANT1* or *XYLT1* genes. This review synthesizes the findings of 111 published case reports, including 54 cases of DBQD1, 39 cases of DBQD2, and 14 cases of the Kim variant (DDKV). Patients in this cohort had a median birth weight of 2505 g, a median length of 40 cm, and a median occipitofrontal circumference of 33 cm. The review highlights the phenotypic variations across Desbuquois dysplasia subtypes, particularly in facial characteristics, joint dislocations, and bone deformities. Genetic analyses revealed a considerable diversity in mutations, with over 35% of cases involving missense mutations, primarily affecting the *CANT1* gene. Additionally, approximately 60% of patients had a history of parental consanguinity, indicating a potential genetic predisposition in certain populations. The identified mutations included deletions, insertions, and nucleotide substitutions, many of which resulted in premature stop codons and the production of truncated, likely nonfunctional proteins. These findings underscore the genetic and clinical complexity of Desbuquois dysplasia, highlighting the importance of early diagnosis and the potential for personalized therapeutic approaches. Continued research is essential to uncover the underlying mechanisms of this disorder and improve outcomes for affected individuals through targeted treatments.

## 1. Introduction

In the field of the genetic classification of skeletal disorders, over 700 distinct disease entities have been identified, many of which are characterized by short stature. The majority of these disorders are rare, with limited knowledge available, and publications often describe only individual cases. The restricted understanding of these rare conditions that are associated with short stature, coupled with the limited availability of genetic testing, can lead to challenges in ordering appropriate investigations and making accurate diagnoses in infants by physicians. Although short stature is one of the most common reasons for referral to growth specialists, only a small percentage of children with this condition receive a molecular diagnosis [1,2]. One of the rare disorders of this type is Desbuquois syndrome (DBQD). This condition belongs to the group of osteochondrodysplasias and is characterized by severe dwarfism with micromelia, facial dysmorphism, joint hypermobility with multiple dislocations, vertebral and metaphyseal anomalies, and accelerated ossification of the carpal and tarsal bones. Two types of this disorder are distinguished based on the presence (type 1, MIM 251450) or absence (type 2, MIM 615777) of characteristic hand deformities. There is also a variant of DBQD, known as the Kim variant (DDKV), which is characterized by short stature, mild joint anomalies, minor facial deformities, and distinct hand abnormalities. According to the International Nosology of Genetic Skeletal Disorders, DBQD is classified as one of the disorders associated with multiple dislocations. Desbuquois dysplasia type 1 (DBQD1) results from homozygous or compound heterozygous mutations in the calcium-activated nucleotidase 1 (*CANT1*) gene. *CANT1* is an extracellular protein that functions as a tri- and diphosphate nucleotide hydrolase, with the primary role of hydrolyzing uridine diphosphate (UDP), guanosine diphosphate (GDP), uridine triphosphate (UTP), and adenosine diphosphate (ADP). Type 2 results from mutations in the *XYLT1* gene on chromosome 16p12. Type 2 can also result from mutations in the *CANT1* gene, meaning that the presence or absence of hand anomalies does not definitively determine the molecular cause of the disease. *XYLT1* is responsible for the initial step in glycosaminoglycan (GAG) biosynthesis by transferring a xylose residue from UDP-xylose to the serine residues on the core proteins of proteoglycans (PGs), essentially initiating the formation of GAG chains. *CANT1*, on the other hand, plays a crucial role in maintaining the nucleotide balance necessary for glycosylation processes by hydrolyzing UDP, a byproduct of glycosyltransferase reactions, within the endoplasmic reticulum (ER) and Golgi apparatus. This hydrolysis prevents the accumulation of UDP, which could inhibit glycosyltransferase activity, thereby ensuring the smooth continuation of GAG chain elongation. The dysfunction of *CANT1* due to mutations can lead to an impaired ability to remove UDP, which in turn may disrupt the glycosylation process, including GAG elongation and PG maturation. Given that *XYLT1* initiates GAG synthesis and *CANT1* ensures the continuation of this process by managing nucleotide levels, these two proteins may indeed be functionally linked within the same biosynthetic pathway. Their roles in GAG biosynthesis and PG metabolism suggest that mutations in either gene could lead to similar disruptions in the ECM integrity, particularly in cartilage, which is heavily reliant on GAGs and PGs for its structure and function [3,4]. Since 1966, numerous case reports of patients with DBQD have been published in the literature, but a comprehensive review in this area is lacking. Therefore, the aim of this article is to analyze the published case reports of patients with DBQD, focusing on the clinical symptoms and genetic mutations in this patient group. This review also aims to provide recommendations for future research in this field. International databases, such as Thomson (Web of Knowledge), PubMed/Medline, ScienceDirect, Scopus, and Google Scholar, were searched to find published case reports using the terms “Desbuquois dysplasia,” “DBQD,” or “Desbuquois osteochondrodysplasia” up to the year 2023. The analyzed data included the year of publication, gender, age, height, genetic mutations, radiological parameters, and clinical symptoms. All the symptoms reported by the authors in the published case descriptions were documented.

## 2. Case Reports

This manuscript reviews 111 published case reports, with the majority (*n* = 54) in patients diagnosed with DBQD1, followed by DBQD2 (*n* = 39), and DDKV (*n* = 14), according to the available data. Based on the available data, the median birth weight was 2505 g, the median birth length was 40 cm, and the median birth OFC (occipitofrontal circumference) was 33 cm (Table 1). Developmental delay or intellectual disability were reported in 20 patients, while motor delay was noted in 15 patients. Among the 81 patients for whom gender was specified, 49 were male and 32 were female.

## 3. Clinical Characteristics in Desbuquois Dysplasia

The table below (Table 2) presents a summary of the symptom occurrence by dysplasia type, as well as across all the case reports included in the analysis. In DBQD1, the most frequently observed symptoms include a round/flat face (HP:0000311/HP:0012368), midface retrusion (HP:0011800), proptosis (HP:0000520), a depressed nasal bridge (HP:0005280), micrognathia (HP:0000347), a short neck (HP:0000470), a narrow chest (HP: 0000774), talipes equinovarus (HP:0001762), a broad thumb (HP:0011304), hypotonia (HP:0001252), and clinodactyly (HP:0030084). In DBQD2, compared with DBQD1, a higher percentage of patients exhibit a flat face (HP:0012368), proptosis (HP:0000520), and a cleft palate (HP:0000175). Among patients with DDKV dysplasia, a significantly greater proportion, compared with DBQD1 and DBQD2, present with a round face (HP:0000311), proptosis (HP:0000520), and a depressed nasal bridge (HP:0005280). What distinguishes DDKV dysplasia is the high percentage of patients with hypertelorism (HP:0000316), elongated fingers (HP:0100807), genu varum (HP:0002970), pes planus (HP:0001763), a short hallux (HP:0010109), and a wide sandal gap (HP:0001852).

Regarding joint dislocation, a higher percentage of patients with the different types of dysplasia exhibited joint laxity/dislocation (HP:0001382/HP:0001373). Notably, in patients with DDKV dysplasia, there was a higher percentage of dislocations of the knee (HP:0004976) and phalangeal dislocations (HP:0006243). The data presented in the table below also indicate that osteoporosis/osteopenia was observed in a greater percentage of patients with DBQD1 and DBQD2 dysplasia types compared with DDKV. This suggests that the observed osteoporosis in patients with DBQD1 and DBQD2 may have significant clinical implications, as it is well known that this condition can lead to low-trauma fractures. Additionally, nearly all patients were found to have the characteristic prominent lesser trochanter (HP:6000816, often called ‘Swedish key’ or described as a ‘monkey wrench’ appearance). A higher percentage of patients with DDKV dysplasia exhibited an elevated greater trochanter, while this feature was observed in only three patients with DBQD1 dysplasia. A short femoral neck (HP:0100864) was primarily characteristic of patients with DBQD2 dysplasia. Among all the analyzed case reports, nearly 80% showed advanced carpal bone age (HP:0004233). Particularly noteworthy is the accessory ossification of the proximal phalanx, which is typical only in DBQD1 dysplasia. The same applies to the bifid distal phalanx of the thumb (HP:0009611) and triangular epiphysis of the phalanx of the hand (HP:0010238, often called delta phalanx). Over 60% of patients with DDKV dysplasia had phalangeal dislocations, while for the other dysplasia types, these were only sporadically reported. Concerning long bones, a lower percentage of patients with DBQD1 dysplasia exhibited metaphyseal widening (HP:0003016), whereas this feature was more common in the other dysplasia types. The shortening of tubular bones was a typical feature of DBQD2 dysplasia, while premature degenerative spondylosis was characteristic of DDKV dysplasia.

## 4. Molecular Genetics

Genetic mutation data in the group of patients with Desbuquois dysplasia began to be published in 2009 (Table 3). Among the genetic mutation data available for 65 case reports, over 35% were missense mutations. Nearly 70% of these mutations involved the *CANT1* gene. According to the available publication data, parental consanguinity was present in about 60% of patients. Mutations associated with Desbuquois dysplasia exhibit significant diversity, including deletions, insertions, and nucleotide substitutions. These mutations occur in various regions of the genes, including exons (Ex1-Ex10) and introns (In1-In7). Many of these mutations result in the creation of premature stop codons, leading to truncated and likely nonfunctional proteins, such as p.Trp125Ter, p.Tyr178LeufsTer4, and p.Arg598AlafsTer7. Although the identified premature termination codons in Desbuquois dysplasia are likely to lead to nonsense-mediated mRNA decay (NMD), resulting in the absence of truncated protein products, there is currently no direct evidence in the literature that is specific to this disease, confirming whether NMD is the predominant outcome or if any truncated proteins are indeed produced. Frameshift mutations, which drastically alter the protein’s structure and function, are also common. Additionally, substitutions that change single amino acids, such as p.Arg300Cys, p.Arg300His, and p.Val226Met, have been identified, potentially affecting protein conformation and activity. These mutations highlight significant disruptions in the genes that are critical for proper bone development, which is characteristic of Desbuquois dysplasia.

## 5. Discussion

To the best of our knowledge, this is the first comprehensive review summarizing published case reports of patients with Desbuquois dysplasia. A total of 111 case reports on this condition were included in this review. The analysis of genetic mutations in patients with Desbuquois dysplasia reveals significant variability in both the types of mutations and their locations within the genes. Among the available case reports, missense mutations account for over 35% of all mutations, highlighting their importance in the pathogenesis of this dysplasia. Such mutations, involving the substitution of a single amino acid, can have diverse effects at the protein level, ranging from subtle conformational changes to significant disruptions in protein function. In the context of Desbuquois dysplasia, where proper protein function is crucial for bone development, even minor structural changes can lead to serious clinical consequences.

Mutations in the *CANT1* gene play a pivotal role in the pathogenesis of Desbuquois dysplasia, which is also reflected in their significance in various types of cancers, such as lung and kidney cancer. The high frequency of missense mutations in *CANT1* in patients with Desbuquois dysplasia suggests that disruptions in this gene’s function may lead to significant alterations in bone development, similar to how they influence cancer progression through signaling pathways, such as NF-κB. Understanding the role of *CANT1* in cancers may provide valuable insights for further research into its role in Desbuquois dysplasia, particularly in the context of the tissue microenvironment and cellular response. Although the mechanisms in Desbuquois dysplasia differ from those observed in oncology, the potential of *CANT1* as a biomarker suggests that it may also be significant in the diagnosis or prognosis of this condition. Further research is needed to more precisely determine how mutations in *CANT1* affect bone tissue development and function in Desbuquois dysplasia. Thus, experience from studies on *CANT1* in cancer may serve as a valuable reference point in the context of genetic disorders such as Desbuquois dysplasia [49,50].

A review of the available data from 111 published case reports of patients with Desbuquois dysplasia indicates a predominance of DBQD1 cases, which may suggest a higher prevalence of this form of the disease in the population or a greater ease in diagnosing it compared with DBQD2 and DDKV. The observed developmental delays, including motor and intellectual delays, affecting a significant number of patients, underscore the need for further research on the pathogenesis of these disorders in the context of Desbuquois dysplasia, as well as the necessity for early therapeutic intervention. The analysis of clinical symptoms across different types of Desbuquois dysplasia (DBQD1, DBQD2, DDKV) reveals significant phenotypic differences that have both diagnostic and prognostic implications. These variants differ in the frequency of certain features, suggesting that they may arise from distinct genetic mechanisms or developmental processes. Particularly noteworthy is the variability in bone deformities and joint characteristics, which may impact disease progression and necessitate a more individualized therapeutic approach. In studies on achondroplasia, such as CLARITY, specific risk factors—such as the presence of hydrocephalus requiring shunt placement or the need for cervicomedullary decompression—were found to significantly increase the likelihood of surgical intervention. These findings underscore the importance of identifying individual risk factors when tailoring personalized therapeutic strategies, ensuring that treatment is optimized for each patient’s specific clinical profile [51]. A similar approach can be applied in the treatment of patients with Desbuquois dysplasia, where phenotypic differences, such as the presence of joint dislocations or specific bone deformities, may require adjustments in therapeutic strategies. Current clinical therapies for patients with skeletal dysplasias, including achondroplasia, are predominantly palliative, although enzyme replacement therapies have been introduced for certain conditions in recent years. The success of these therapies is based on an accurate molecular diagnosis and a thorough understanding of the pathogenic pathways that affect bone growth and development. In achondroplasia, a personalized therapeutic approach, grounded in the precise identification of mutations and their associated pathogenic mechanisms, is becoming increasingly feasible due to the advancements in molecular technology [52]. In the case of Desbuquois dysplasia, a similar approach could yield significant benefits. Although targeted therapies for Desbuquois dysplasia are currently lacking, future research should focus on identifying specific biomarkers and understanding the pathogenic pathways involved. Such knowledge could enable the development of personalized treatment methods that are more effective and tailored to the individual needs of patients. Advances in molecular diagnostics and genetic technology offer the potential for an earlier and more accurate diagnosis of Desbuquois dysplasia, which in the future could pave the way for the introduction of gene therapies or other causative treatments. Therefore, in the coming years, it is crucial to conduct research on Desbuquois dysplasia that may contribute to the personalization of medical care and improve treatment outcomes in these patients. The management of patients with Desbuquois dysplasia requires a multidisciplinary approach tailored to the specific clinical manifestations and the severity of the condition. A regular physical therapy program is recommended to address joint laxity and to prevent or delay the onset of early-onset osteoarthritis, which is a common complication. Surgical interventions may be necessary for severe joint dislocations, spinal deformities, or other significant skeletal abnormalities [45,53]. Additionally, careful airway management is crucial, particularly in cases where craniofacial abnormalities complicate tracheal intubation. The use of supraglottic devices, like the CobraPLA, may provide a viable alternative for maintaining a secure airway. However, the success of such devices may vary, and backup plans, including the use of advanced intubation techniques, should be prepared [54].

This review highlights the significant variability in genetic mutations associated with Desbuquois dysplasia and their impact on clinical phenotypes. The predominance of DBQD1 cases in the literature suggests that further studies should investigate whether this reflects a true higher prevalence or whether it is a consequence of diagnostic biases. Additionally, the observation of developmental delays in a significant proportion of patients underscores the importance of early diagnosis and intervention. Future research should focus on several key areas: firstly, further studies are needed to deepen our understanding of the correlation between specific mutations and the clinical manifestations of Desbuquois dysplasia, which could help to refine the diagnostic criteria and improve prognostic predictions. Secondly, investigating the pathogenic pathways involving *CANT1* and other relevant genes in Desbuquois dysplasia, particularly in comparison with their roles in other conditions, such as cancers, could uncover new insights into the disease’s underlying mechanisms and identify potential biomarkers for earlier diagnosis or predicting disease progression. Thirdly, given the phenotypic variability observed among patients, there is a need for personalized therapeutic strategies that are tailored to individual clinical profiles. Future research should explore the potential for developing targeted therapies, such as enzyme replacement or gene therapies, based on the molecular and genetic characteristics of each patient. Moreover, conducting longitudinal studies in patients with Desbuquois dysplasia would provide valuable data on disease progression, response to treatment, and long-term outcomes, helping to identify critical periods for intervention that might improve patient outcomes. Finally, the development and implementation of advanced molecular diagnostic techniques could facilitate an earlier and more accurate diagnosis of Desbuquois dysplasia, allowing for early intervention and potentially improving the quality of life and prognosis in affected individuals. While significant progress has been made in understanding Desbuquois dysplasia, ongoing research is essential to translate these findings into improved clinical care. The integration of molecular genetics into clinical practice holds the promise of more effective and individualized treatments for patients with this rare and challenging condition.

## Figures and Tables

**Table 1 ijms-25-09700-t001:** Gender distribution, birth weight, birth length, and birth OFC (cm) in the case reports of patients with Desbuquois syndrome.

Reference	Gender	Birth Weight (g)	Birth Length (cm)	Birth OFC (cm)	Type
[5]	F	2100			DBQD1
	F	2200			DBQD1
[6]	M	2260			DBQD1
	M	2600	38		DBQD1
	M	2700			DBQD1
[7]	F	3000	40		DBQD2
	F		40		DBQD2
[8]	M	2880			DBQD2
[9]	F				DBQD2
	F				DBQD2
[10]	F	3370	42	34	DBQD2
[11]	M	2100	43		DBQD1
	M	2850	41.5	33	DBQD2
	F	2630	41	32.5	DBQD2
	M				DBQD2
	M				DBQD2
[12]	M	1325	32	30.5	DBQD1
[13]	F	2490	35		DBQD2
		3300			DBQD2
[14]	F				DBQD1
	F				DBQD1
	M				DBQD1
	F				DBQD1
	M				DBQD1
	F	2270	37	33	DBQD1
	F	2255			DBQD1
[15]	F	2880	41	34	DBQD2
	M	3400	42	36	DBQD2
	M	1900	37	32.5	DBQD1
[16]	M	2250			DBQD1
[17]	M	2730	49.3	32.5	DBQD2
[18]	F	2820	48		DBQD2
	M	3200			DBQD2
		3000			DBQD2
[19]	M				DBQD1
	M				DBQD1
	M				DBQD1
[20]	F	2900	40		DBQD2
			37		DBQD2
[21]	M		42		DBQD2
[22]	F	2400	44	35	
	M	3100	49	37	
	F				
[23]	F	1055	30	25.5	DBQD1
[24]	F				DBQD1
[25]	M				DBQD1
[26]	F		34		DBQD1
	M		33		DBQD1
	M		34		DBQD1
	F				DBQD1
	M				DBQD1
	M		43		DBQD1
	M				DBQD1
	M		37		DBQD1
	M		36		DBQD1
	M		35		DBQD1
[27]	M	2100	36.5	32	DBQD1
[28]	M	3400			DDKV
	M	3200			DDKV
	M	3200	46		DDKV
	F	3500	52		DDKV
	F	2100			DDKV
	M	2400			DDKV
	M				DDKV
[29]					DBQD1
					DBQD2
					DBQD2
					DDKV
[30]	M				DBQD1
	M				DBQD1
	M				DBQD1
[31]					DBQD1
			41		DBQD1
					DBQD1
					DBQD1
					DBQD1
					DBQD1
					DDKV
					DBQD2 with atypical hand anomalies
[32]	F		37		DBQD2
	M		41		DBQD2
	F	2000			DBQD2
	M	2570	39	33	DBQD2
			44		DBQD2
			43		DBQD2
		1200	33		DBQD2
[33]	M	1687	31	32.3	DBQD1
[34]	M				DDKV
	M				DDKV
	F				DDKV
[35]	M	2930	45	37	DBQD2
[36]	F	2500			DBQD2
[37]		2510	40.5		DBQD2
[38]	M	2440	38	33	DBQD2
[39]	F	2600			DBQD2
		1800			DBQD2
		1700	36	31	DBQD2
[40]	F				DBQD1
[41]	M				DBQD1
[42]					DBQD1
					DBQD1
					DBQD1
[43]					DDKV
[44]					DBQD1
[45]	M				DDKV
[46]	F	1850			DBQD1
	M	1000			DBQD1
[47]					DBQD2
[48]					DBQD1
					DBQD1

**Table 2 ijms-25-09700-t002:** Summary of symptom occurrence by dysplasia type and across all the case reports included in the analysis.

	Type of Dysplasia			
	DBQD1 n (%)	DBQD2 n (%)	DDKV n (%)	In Total n (%)
Antenatal case	15 (28%)	1 (3%)	0 (0%)	16 (15%)
Early death (shortened life expectancy)	15 (28%)	2 (5%)	0 (0%)	17 (16%)
RDS/respiratory failure/respiratory arrest	14 (26%)	10 (26%)	0 (0%)	24 (22%)
**Dysmorphic Features and Associated Clinical Findings**				
Narrow chest (HP:0000774)	27 (50%)	21 (54%)	1 (7%)	49 (46%)
Depressed nasal bridge (HP:0005280)	20 (37%)	17 (44%)	9 (64%)	46 (43%)
Proptosis (HP:0000520)	19 (35%)	18 (46%)	8 (57%)	45 (42%)
Round face (HP:0000311)	20 (37%)	14 (36%)	10 (71%)	44 (41%)
Short neck (HP:0000470)	25 (46%)	13 (33%)	2 (14%)	40 (37%)
Flat face (HP:0012368)	19 (35%)	19 (49%)	1 (7%)	39 (36%)
Micrognathia (HP:0000347)	23 (43%)	9 (23%)	0 (0%)	32 (30%)
Midface retrusion (HP:0011800)	18 (33%)	9 (23%)	3 (21%)	30 (28%)
Hypotonia (HP:0001252)	15 (28%)	4 (10%)	4 (29%)	23 (21%)
Talipes equinovarus (HP:0001762)	15 (28%)	2 (5%)	5 (36%)	22 (21%)
Scoliosis (HP:0002650)	8 (15%)	6 (15%)	7 (50%)	21 (20%)
Genu varum (HP:0002970)	4 (7%)	7 (18%)	9 (64%)	20 (19%)
Pes planus (HP:0001763)	3 (6%)	6 (15%)	8 (57%)	17 (16%)
Clinodactyly (HP:0030084)	17 (31%)	0 (0%)	0 (0%)	17 (16%)
Long philtrum (HP:0000343)	9 (17%)	7 (18%)	0 (0%)	16 (15%)
Cleft palate (HP:0000175)	6 (11%)	10 (26%)	0 (0%)	16 (15%)
Hypertelorism (HP:0000316)	2 (4%)	6 (15%)	7 (50%)	15 (14%)
Broad thumb (HP:0011304)	14 (26%)	1 (3%)	0 (0%)	15 (14%)
Anteverted nares (HP:0000463)	6 (11%)	7 (18%)	0 (0%)	13 (12%)
Blue sclerae (HP:0000592)	3 (6%)	7 (18%)	1 (7%)	11 (10%)
Retrognathia (HP:0000278)	7 (13%)	4 (10%)	0 (0%)	11 (10%)
Flexion contracture of the digit (HP:0030044)	9 (17%)	1 (3%)	1 (7%)	11 (10%)
Short hallux (HP:0010109)	2 (4%)	0 (0%)	9 (64%)	11 (10%)
Short nose (HP:0003196)	6 (11%)	4 (10%)	0 (0%)	10 (9%)
Short finger (HP:0009381)	5 (9%)	4 (10%)	1 (7%)	10 (9%)
Sandal gap (HP:0001852)	2 (4%)	0 (0%)	8 (57%)	10 (9%)
Metatarsus adductus (HP:0001840)	2 (4%)	1 (3%)	7 (50%)	10 (9%)
Narrow mouth (HP:0000160)	8 (15%)	1 (3%)	0 (0%)	9 (8%)
Abdominal distention (HP:0003270)	5 (9%)	4 (10%)	0 (0%)	9 (8%)
Hyperlordosis (HP:0003307)	4 (7%)	5 (13%)	0 (0%)	9 (8%)
Genu valgum (HP:0002857)	3 (6%)	6 (15%)	0 (0%)	9 (8%)
Smooth philtrum (HP:0000319)	7 (13%)	1 (3%)	0 (0%)	8 (7%)
Concave nasal ridge (HP:0011120)	7 (13%)	1 (3%)	0 (0%)	8 (7%)
Long fingers (HP:0100807)	0 (0%)	0 (0%)	8 (57%)	8 (7%)
Obesity (HP:0001513)	2 (4%)	5 (13%)	1 (7%)	8 (7%)
Brachycephaly (HP:0000248)	7 (13%)	1 (3%)	0 (0%)	8 (7%)
Brachydactyly (HP:0001156)	4 (7%)	3 (8%)	0 (0%)	7 (7%)
Proximal placement of the thumb (HP:0009623)	7 (13%)	0 (0%)	0 (0%)	7 (7%)
**Joint dislocation**				
Joint hypermobility/dislocation (HP:0001382/HP:0001373)	43 (80%)	29 (74%)	13 (93%)	85 (79%)
Dislocation of the knee (HP:0004976)	18 (33%)	8 (21%)	8 (57%)	34 (32%)
Hip dislocation (HP:0002827)	15 (28%)	8 (21%)	2 (14%)	25 (23%)
Phalangeal dislocation (HP:0006243)	11 (20%)	1 (3%)	9 (64%)	21 (20%)
Elbow dislocation (HP:0003042)	11 (20%)	3 (8%)	0 (0%)	14 (13%)
Patellar dislocation (HP:0002999)	8 (15%)	2 (5%)	1 (7%)	11 (10%)
**Fetal clinical findings**				
Hydrops fetalis (HP:0001789)	4 (7%)	0 (0%)	0 (0%)	4 (4%)
Increased nuchal translucency (HP:0010880)	4 (7%)	0 (0%)	0 (0%)	4 (4%)
Polyhydramnios (HP:0001561)	3 (6%)	0 (0%)	0 (0%)	3 (3%)
**Radiological features**				
Osteoporosis/Osteopenia (HP:0000939/HP:0000938)	9 (17%)	17 (44%)	2 (14%)	28 (26%)
**Pelvis and hips**				
Prominent lesser trochanter (HP:6000816) (‘Swedish key’/‘monkey wrench’ appearance)	50 (93%)	35 (90%)	14 (100%)	99 (93%)
Flat acetabular roof (HP:0003180)	19 (35%)	10 (26%)	2 (14%)	31 (29%)
Short femoral neck (HP:0100864)	4 (7%)	11 (28%)	2 (14%)	17 (16%)
Elevated greater trochanter	3 (6%)	0 (0%)	9 (64%)	12 (11%)
Hip joint space narrowing	0 (0%)	0 (0%)	12 (86%)	12 (11%)
Beaking of femur neck/metaphysis	8 (15%)	2 (5%)	1 (7%)	11 (10%)
Coxa vara (HP:0002812)	3 (6%)	2 (5%)	4 (29%)	9 (8%)
Flared iliac wing (HP:0002869)	7 (13%)	1 (3%)	1 (7%)	9 (8%)
Coxa valga (HP:0002673)	1 (2%)	1 (3%)	5 (36%)	7 (7%)
Broad femoral neck (HP:0006429)	2 (4%)	3 (8%)	1 (7%)	6 (6%)
**Hands**				
Advanced carpal bone age (HP:0004233)	39 (72%)	33 (85%)	12 (86%)	84 (79%)
Accessory ossification proximal phalanx *(between the metacarpal and proximal phalanx)*, 2nd digit (index)	32 (59%)	0 (0%)	0 (0%)	32 (30%)
Short metacarpals (no other info) (HP:0010049)	8 (15%)	10 (26%)	8 (57%)	26 (24%)
Triangular epiphyses of the phalanges of the hand (HP:0010238) (Delta phalanx/delta-like phalanx)	20 (37%)	0 (0%)	0 (0%)	20 (19%)
Bifid distal phalanx of the thumb (HP:0009611)	14 (26%)	0 (0%)	0 (0%)	14 (13%)
Short 1st metacarpals (HP:0010034)	7 (13%)	1 (3%)	6 (43%)	14 (13%)
Short distal phalanx of the finger (HP:0009882)	0 (0%)	1 (3%)	12 (86%)	13 (12%)
Radial deviation of the 2nd finger (HP:0009467)	11 (20%)	0 (0%)	0 (0%)	11 (10%)
Premature fusion of phalangeal epiphyses (HP:0006140)	0 (0%)	1 (3%)	8 (57%)	9 (8%)
Deviation of fingers (HP:0004097)	8 (15%)	0 (0%)	0 (0%)	8 (7%)
Narrowing/fusion of the intercarpal space (HP:0009702)	1 (2%)	0 (0%)	7 (50%)	8 (7%)
Shortening of all metacarpals (HP:0005720)	0 (0%)	3 (8%)	4 (29%)	7 (7%)
Short phalanges (HP:0009803)	0 (0%)	7 (18%)	0 (0%)	7 (7%)
Elongated proximal phalanges	0 (0%)	0 (0%)	6 (43%)	6 (6%)
Elongated middle phalanges	0 (0%)	0 (0%)	6 (43%)	6 (6%)
Metacarpophalangeal (mp) joint dislocation	6 (11%)	0 (0%)	0 (0%)	6 (6%)
Accessory ossification proximal phalanx, 3rd digit	5 (9%)	0 (0%)	0 (0%)	5 (5%)
**Feet**				
Advanced tarsal ossification (HP:0008108)	11 (20%)	12 (31%)	3 (21%)	26 (24%)
Short metatarsal (HP:0010743)	8 (15%)	2 (5%)	8 (57%)	18 (17%)
Extraneous ossification centers	8 (15%)	2 (5%)	0 (0%)	10 (9%)
Triangular epiphyses of the toes (HP:0010172) (Delta phalanx/delta-like phalanx)	9 (17%)	0 (0%)	0 (0%)	9 (8%)
Hallux valgus (HP:0001822)	8 (15%)	1 (3%)	0 (0%)	9 (8%)
Distal fibular overgrowth	7 (13%)	0 (0%)	1 (7%)	8 (7%)
**Long bones**				
Metaphyseal widening (HP:0003016)	10 (19%)	18 (46%)	7 (50%)	35 (33%)
Shortening of tubular bones	11 (20%)	18 (46%)	0 (0%)	29 (27%)
Flared metaphysis (HP:0003015)	14 (26%)	6 (15%)	0 (0%)	20 (19%)
Proximal fibular overgrowth (HP:0005067)	8 (15%)	0 (0%)	8 (57%)	16 (15%)
Abnormal epiphysis morphology (HP:0005930)	1 (2%)	7 (18%)	3 (21%)	11 (10%)
Flat metaphyses	1 (2%)	1 (3%)	7 (50%)	9 (8%)
Flattened epiphysis (HP:0003071)	1 (2%)	5 (13%)	3 (21%)	9 (8%)
Enlarged metaphyses (HP:0003051)	2 (4%)	4 (10%)	1 (7%)	7 (7%)
Short diaphyses (HP:0000941)	7 (13%)	0 (0%)	0 (0%)	7 (7%)
Bowing of the long bones (HP:0006487)	6 (11%)	1 (3%)	0 (0%)	7 (7%)
**Spine and thorax**				
Coronal cleft vertebrae (HP:0003417)	15 (28%)	7 (18%)	1 (7%)	23 (21%)
Intervertebral space narrowing (HP:0002945)	3 (6%)	0 (0%)	11 (79%)	14 (13%)
Platyspondyly (HP:0000926)	3 (6%)	9 (23%)	1 (7%)	13 (12%)
Abnormally ossified vertebrae (HP:0100569)	5 (9%)	7 (18%)	0 (0%)	12 (11%)
Abnormality of the vertebral endplates (HP:0005106)	0 (0%)	2 (5%)	8 (57%)	10 (9%)
Abnormality of the cervical spine (HP:0003319)	8 (15%)	1 (3%)	0 (0%)	9 (8%)
Premature degenerative spondylosis	0 (0%)	0 (0%)	9 (64%)	9 (8%)
Increased vertebral height (HP:0004570)	8 (15%)	0 (0%)	0 (0%)	8 (7%)

**Table 3 ijms-25-09700-t003:** Genetic mutation data in patients with Desbuquois dysplasia.

Reference	Ethicity	Parental Consanguinity (Y–Yes, N–No)	Gene	Location	Nucleotide Change ^†^	Amino Acid Change/Effect on Protein	Type of Mutation	Type
[26]	Sri Lankan	Y	*CANT1*	50 UTR and Ex1	del 2703 bp (hom ^‡^)		nonsense	DBQD1
	Turkish	Y	*CANT1*	Ex3	c.734 delC (hom)	p.Pro245ArgfsTer3	nonsense	DBQD1
	Turkish	Y	*CANT1*	Ex4	c.898C>T (hom)	p.Arg300Cys	missense	DBQD1
	Turkish	Y	*CANT1*	Ex4	c.898C>T (hom)	p.Arg300Cys	missense	DBQD1
	Iranian	Y	*CANT1*	Ex4	c.898C>T (hom)	p.Arg300Cys	missense	DBQD1
	French	Y	*CANT1*	Ex4	c.899G>A (hom)	p.Arg300His	missense	DBQD1
	Emirati	Y	*CANT1*	Ex4	c.899G>A (hom)	p.Arg300His	missense	DBQD1
	Moroccan	Y	*CANT1*	Ex4	c.907-911insGCGCC (hom)	p.Ser303AlafsTer20	frameshift	DBQD1
	Brazilian	Y	*CANT1*	Ex2	c.374G>A (hom)	p.Trp125Ter	nonsense	DBQD1
	Brazilian	Y	*CANT1*	Ex4	c.896C>T (hom)	p.Pro299Leu	missense	DBQD1
[27]	Saudi	Y	*CANT1*	Ex4	NM_001159772.1:c.893-894insGCCGC (hom)	p.325fsTer	frameshift	DBQD1
[28]	Korean	N	*CANT1*	In2/Ex3	IVS2–9G>A/c.676G>A (chet ^§^)	p.Gly279ValfsTer8/p.Val226Met	(mutation at splice acceptor site)/missense	DDKV
	Korean	N						DDKV
	Japanese	Y	*CANT1*	Ex3	c.676G>A (hom)	p.Val226Met	missense	DDKV
	Japanese	Y						DDKV
	Japanese	N	*CANT1*	Ex3/Ex4	c.676G>A/c.861C>A (chet)	p.Val226Met/p.Cys287Ter	missense/nonsense	DDKV
	Korean	N	*CANT1*	Ex3/Ex4	c.676G>A/c.1079C>A (chet)	p.Val226Met/p.Ala360Asp	missense/missense	DDKV
	Turkish	N						DDKV
[29]	Australian	N	*CANT1*	Ex2	c.228_229insC/c.617T>C (chet)	p.Trp77LeufsTer13/p.Leu224Pro	nonsense/missense	DBQD1
	Turkish	Y	*CANT1*	Ex2	c.375G>C (hom)	p.Trp125Cys	missense	DBQD2
	Turkish	Y						DBQD2
	Japanese	N	*CANT1*		c.676G>A/c.494T>C (chet)	p.Val226Met/p.Met165Thr	missense/missense	DDKV
[30]	German	Y	*CANT1*	Ex2	c.336C>A (hom)	p.Asp112Glu	missense	DBQD1
	German	N	*CANT1*	Ex2	c.277_278delCT/c.228_229insC (chet)	p.Leu93ValfsTer89/p.Trp77LeufsTer13	frameshift/frameshift	DBQD1
	German	N	*CANT1*	Ex2	c.228_229insC (hom)	p.Trp77LeufsTer13	frameshift	DBQD1
[31]	Israeli	Y	*CANT1*	Ex4	c.899G>A (hom)	p.Arg300His	missense	DBQD1
	Moroccan	Y	*CANT1*	Ex4	c.1121T>A (hom)	p.Ile374Asn	missense	DBQD1
	Dutch (Surinamese Hindustan descent)	N	*CANT1*	Ex2	c.100delinsTT/c.358delC (chet)	p.Ala34PhefsTer56/p.Gln120LysfsTer10	frameshift/frameshift	DBQD1
	Turkish	Y	*CANT1*	Ex2	c.531_532del>T (hom)	p.Tyr178LeufsTer4	frameshift	DBQD1
	Yemeni	Y	*CANT1*	Ex2	c.531_532del>T (hom)	p.Tyr178LeufsTer4	frameshift	DBQD1
	Bangladeshi	N	*CANT1*	Ex2	c.277_278delCT/c.100delinsTT (chet)	p.Leu93ValfsTer89/p.Ala34PhefsTer56	frameshift/frameshift	DBQD1
	Turkish	Y	*CANT1*	Ex4	c.909C>G (hom)	p.Ser303Arg	missense	DDKV
	Turkish	Y	*CANT1*	In1	c.–286+1G>A (hom)		intronic splice site mutation	DBQD 2
	Syrian	Y	*CHST3*		NM_004273.5:c.776T>C (hom)	p.Leu259Pro	missense	
[32]	Tunisian	Y	*XYLT1*	Ex9	c.1792C>T (hom)	p.Arg598Cys	missense	DBQD2
	Tunisian	Y	*XYLT1*	Ex9	c.1792C>T (hom)	p.Arg598Cys	missense	DBQD2
	Mauritian	Y	*XYLT1*	Ex3	c.439C>T (hom)	p.Arg147Ter	nonsense	DBQD2
	Belgian	Y	*XYLT1*	Ex1	c.276dupG (hom)	p.Pro93AlafsTer69	frameshift	DBQD2
	Turkish	Y	*XYLT1*	In7	c.1588–3C>T (hom)		splice site mutation	DBQD2
	Turkish	Y	*XYLT1*	In5	c.1290–2A>C (hom)		splice site mutation	DBQD2
	Turkish	Y	*XYLT1*	In5	c.1290–2A>C (hom)		splice site mutation	DBQD2
[33]	Japanese	N	*CANT1*	Ex3	c.805delC (hom)	p.Leu269CysfsTer54	frameshift	DBQD1
[34]	Indian	Y	*CANT1*	Ex2	c.467C>T (hom)	p.Ser156Phe	missense	DDKV
	Indian	Y	*CANT1*	Ex2	c.467C>T (hom)	p.Ser156Phe	missense	DDKV
	Indian	Y	*CANT1*	Ex2	c.467C>T (hom)	p.Ser156Phe	missense	DDKV
[35]	Polish	N	*XYLT1*		c.595C>T/c.1651C>T (chet)	p.Gln199Ter/p.Arg551Cys	nonsense/missense	DBQD2
[36]	Brazilian	Y	*XYLT1*	Ex8	c.1651C > T (hom)	p.Arg551Cys	missense	DBQD2
[37]	Dutch	N	*XYLT1*	Whole gene/In7-Ex8	16p13 del (3.3 Mb)/c.1588-10_1595del (chet)			DBQD2
[38]	Emirati	Y	*XYLT1*	Ex10	c.2169dupA (hom)	p.Val724SerfsTer10	frameshift	DBQD2
[39]	Turkish	Y	*XYLT1*	Ex9	c.1792delC (hom)	p.Arg598AlafsTer7	nonsense	DBQD2
	Turkish	Y	*XYLT1*	Ex9	c.1792delC (hom)	p.Arg598AlafsTer7	nonsense	DBQD2
	Turkish	N	*XYLT1*	In5	c.1290-2A>C (hom)		splice site mutation	DBQD2
[40]	Pakistani	Y	*CANT1*	Ex3	c.643G>T (hom)	p.Glu215Term	nonsense	DBQD1
[41]	Chinese	Y	*CANT1*	In3	NM_138793:c.836-9G>A (hom)	p.Gly279ValfsTer8		DBQD1
[42]	Chinese		*CANT1*		c.594G>A/c.734C>T (chet)	p.Trp198Ter/p.Pro245Leu	nonsense/missense	DBQD1
	Chinese		*CANT1*		c.594G>A/c.734C>T (chet)	p.Trp198Ter/p.Pro245Leu	nonsense/missense	DBQD1
	Chinese		*CANT1*		c.594G>A/c.734C>T (chet)	p.Trp198Ter/p.Pro245Leu	nonsense/missense	DBQD1
[43]	Indian	N	*CANT1*		c.467C>T (hom)	p.Ser156Phe	missense	DDKV
[44]	French	N	*CANT1*		c.340G > A/c.277_278del (chet)	p.Asp114Asn/p.leu93ValfsTer89	missense/frameshift	DBQD1
[45]	Turkish	N	*CANT1*		c.375 G>C (hom)	p. Trp125Cys	missense	DDKV
[46]	Egyptian	Y	*CANT1*	Ex2	NM_001159772.1:c.277_278delCT (hom)	p.Leu93ValfsTer89	frameshift	DBQD1
	Egyptian	Y	*CANT1*	Ex4	NM_001159772.1:c.898C>T (hom)	p.Arg300Cys	missense	DBQD1
[47]	Iranian	N	*XYLT1*		NM_022166.4:c.742G>A/c.1537 C>A (chet)	Glu248Lys/Leu513Met	missense/missense	DBQD2
[48]	Indian	Y	*CANT1*		c.896C>T (hom)	p.Pro299Leu	missense	DBQD1
	Indian	Y	*CANT1*		c.906_907insGCGCC (hom)	p.Ser303AlafsTer20	frameshift	DBQD1

^‡^ hom, homozygous; ^§^ chet, compound heterozygous; ^†^ reference sequence: *CANT1*–NM_001159773.2 if not otherwise specified, *XYLT1*–NM_022166.3 if not otherwise specified.

## Data Availability

No new data were generated.

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
