# Peer review of "Clinical and Genetic Insights into Desbuquois Dysplasia: Review of 111 Case Reports"

_ijms, 2024, doi:10.3390/ijms25179700_

Round 1

Reviewer 1 Report

Comments and Suggestions for Authors

This is a summary of clinical and mutational literature reports in Desbuquoi syndrome. The main purpose of the article is to list clinical findings and mutations in two large tables. Consulting these tables may be found useful by readers who wish a quick overview of the condition.

A few items should be clarified and corrected.

 Specific Comments

1.     Abstract: should clearly state the inheritance pattern and whether mutations are monoallelic or biallelic.

2.     Abstract and Results: is there any evidence of protein production in the presence of premature termination codons in this disease? It would be expected that premature termination codons lead to nonsense-mediated decay of mRNA rather than the formation of truncated proteins.

3.     Introduction: The first sentence should cite the new version of the Skeletal Dysplasia Nosology (Unger et al, 2023) and the fact that there are more than 700 skeletal dysplasias. The reference to the older version of the nosology can be omitted.

4.     Table 2 is far too long and not very informative because most entries have only 1 or 0 patients. The authors need to make the effort to combine entries that denote very similar things, even though the original papers may have used different words. It would be more informative to have a table that summarizes the findings on two pages.

5.     As “osteoporosis” seems to be a common finding, there should be a statement whether low-trauma fractures have been reported.

6.     There should be a definition of what is meant by ‘early death’.

Author Response

Dear Reviewer nr 1,

Comment 1: “This is a summary of clinical and mutational literature reports in Desbuquoi syndrome. The main purpose of the article is to list clinical findings and mutations in two large tables. Consulting these tables may be found useful by readers who wish a quick overview of the condition. A few items should be clarified and corrected.”

Answer 1: We appreciate the positive feedback on our manuscript. Below, we provide responses to the valuable suggestions offered by the reviewer.

Comment 2: “Abstract: should clearly state the inheritance pattern and whether mutations are monoallelic or biallelic.”

Answer 2: We have added the information suggested by the reviewer to the abstract.

Comment 3: “Abstract and Results: is there any evidence of protein production in the presence of premature termination codons in this disease? It would be expected that premature termination codons lead to nonsense-mediated decay of mRNA rather than the formation of truncated proteins.”

Answer 3: The response to this question has been included in the genetic mutations section.

Comment 4: “Introduction: The first sentence should cite the new version of the Skeletal Dysplasia Nosology (Unger et al, 2023) and the fact that there are more than 700 skeletal dysplasias. The reference to the older version of the nosology can be omitted.”

Answer 4: Thank you for the valid suggestion. We have replaced the reference as indicated and corrected the number from 400 to 700.

Comment 5: “Table 2 is far too long and not very informative because most entries have only 1 or 0 patients. The authors need to make the effort to combine entries that denote very similar things, even though the original papers may have used different words. It would be more informative to have a table that summarizes the findings on two pages.”

 Answer 5: Thank you for your additional advice; we fully agree with it. The symptoms in Table 2 have been arranged from the most common to the least common, and the table has been condensed by removing symptoms observed in fewer than approximately 6-7 patients.

Comment 6: “As “osteoporosis” seems to be a common finding, there should be a statement whether low-trauma fractures have been reported.”

 Answer 6: Thank you for your observation. While data on low-trauma fractures in patients with Desbuquois dysplasia are not observed in the available literature, it is well known that osteoporosis can lead to such fractures. Therefore, it is possible that patients with osteoporosis associated with DBQD1 and DBQD2 may be at an increased risk for low-trauma fractures. We have added this information in section 3.

Comment 7: “There should be a definition of what is meant by ‘early death’.”

 Answer 7: An explanation of the term "early death" has been added in a parenthesis in the table.

Reviewer 2 Report

Comments and Suggestions for Authors

The manuscript ijms-3191188, "Clinical and Genetic Insights into Desbuquois Dysplasia: Review of 111 Case Reports”, reviews the findings from 111 published case reports, including 54 cases of DBQD1, 39 of DBQD2, and 14 of the Kim variant (DDKV) highlighting the genetic and clinical complexity of Desbuquois dysplasia, and the importance of early diagnosis and the potential for personalized therapeutic approaches.

Considering a lack of comprehensive review in this area, this manuscript significantly contributes to the field. Still, it can be improved to be more constructive and helpful for future studies and research.

Minor comments

1.     All the gene names should be in italics throughout the text and tables.

2.     Please add OMIM numbers for each reported condition.

3.     According to the journal, a review should comprise an Abstract, Keywords, Introduction, Relevant Sections, Discussion, Conclusions, and Future Directions. Since the review should be critical and constructive and provide recommendations for future research, I’d invite the authors to add the missing Conclusions and Future Directions, revising what is reported in the Discussion.

4.     In the Introduction, lines 46-48, “DBQD1 results from homozygous or compound heterozygous mutations in the calcium-activated nucleotidase 1 (CANT1) gene.” it would be better to specify DBQD type 1.

5.     In lines 55-57, the authors should add that this review aims also to provide recommendations for future research in this field.

6.     In lines 71,72, “Table 1. Gender distribution, birth weight, birth length, and birth OFC (cm) in case reports of patients with Desbuquois Syndrome.”, please be consistent and either add reference units such as g and cm where missing or leave out at all.

7.     Subsection 3, “Symptom Occurrence by Dysplasia Type”, could be changed to “Clinical characteristics in Desbquois Dysplasia”.

8.     Subsection 4 “Genetic Mutations” could be changed to “Molecular genetics”.

9.     In Table 2 “Radiolgical features” should be corrected.

10.  In Table 2, when summarising the symptoms, did the authors consider if something was not reported because it was not present or could not be assessed? It may be worthwhile to specify it.

11.  Table 2 has an excessive length; several redundant voices could be trimmed or merged.

12.  In Table 2, regarding the description of the symptoms, the authors should specify whether the clinical terms used are HPO terms, and if they are, they should also add the appropriate code. I’d recommend using this standardised vocabulary since it will significantly help reduce redundancy (i.e. thin vermilion border/thin lips/thin upper lips;  Small, narrow or bell-shaped thorax/Barrel-shaped thorax; various clinodactily in Table 2). In my opinion, it is an effort that will pay to support future studies, improving biomedical research because it standardises the way human phenotypes are described, ensuring consistency and accuracy across different clinical databases and research studies. Therefore, it becomes easier to analyse and interpret clinical data, leading to more accurate and early therapeutic intervention diagnoses and a better understanding of genotype/phenotype correlations.

13.  In Table 3, it would be better to put for consanguinity Y(yes) or N (No), since it is more immediate.

14.  In Table 3, "Ethicity" should be corrected and the relative term should be adjusted (i.e Moroccan not Morocco, Turkish not Turkey, Iranian not Iran…)

15.  According to HGVS Nomenclature by Den Dunnen et al. (2016), it is recommended to use three-letter amino acid code abbreviations for protein variant descriptions; furthermore, so doing they will be concordant throughout the table and the text. Moreover, all variants should be described with an accepted reference sequence, i.e. NMXXXX. The authors could specify this in a note at the end of the table.

16.  I suggest adding a column in Table 3 specifying the mutation’s status, i.e., homozygous or compound heterozygous.

17.  The authors should add a section as “Molecular pathogenesis” and add something about the protein coded by XYLT1 and its functional role. Are the two proteins, XYLT1  and CANT1, somehow implicated in the same pathway? Are there any connections? They should also add something about the physiopathological mechanisms. Is there a connection between the two proteins, XYLT1 and CANT1, through their roles in glycosaminoglycan (GAG) biosynthesis and proteoglycan metabolism? This information will contribute to a better understanding of the underlying physiopathological mechanisms, making the review more comprehensive and helpful.

18.  A discussion about differential diagnosis and management would be interesting to add in the light of a more comprehensive review. The text could also be changed to be more extensive.

Author Response

Dear Reviewer nr 2,

Comment 1: “The manuscript ijms-3191188, "Clinical and Genetic Insights into Desbuquois Dysplasia: Review of 111 Case Reports”, reviews the findings from 111 published case reports, including 54 cases of DBQD1, 39 of DBQD2, and 14 of the Kim variant (DDKV) highlighting the genetic and clinical complexity of Desbuquois dysplasia, and the importance of early diagnosis and the potential for personalized therapeutic approaches. Considering a lack of comprehensive review in this area, this manuscript significantly contributes to the field. Still, it can be improved to be more constructive and helpful for future studies and research.”

Answer 1: Thank you for the positive feedback regarding our manuscript and its significance.

Comment 2: “All the gene names should be in italics throughout the text and tables.”

Answer 2: Thank you for the valuable suggestion. All such terms have been italicized.

Comment 3: “Please add OMIM numbers for each reported condition.”

Answer 3: In the introduction, OMIM codes have been added for the relevant diseases.

Comment 4: “According to the journal, a review should comprise an Abstract, Keywords, Introduction, Relevant Sections, Discussion, Conclusions, and Future Directions. Since the review should be critical and constructive and provide recommendations for future research, I’d invite the authors to add the missing Conclusions and Future Directions, revising what is reported in the Discussion.”

Answer 4: Thank you for the additional suggestion. We have incorporated the recommendations provided by the reviewer in the discussion section.

Comment 5: “In the Introduction, lines 46-48, “DBQD1 results from homozygous or compound heterozygous mutations in the calcium-activated nucleotidase 1 (CANT1) gene.” it would be better to specify DBQD type 1.”

Answer 5: The specified sentence has been corrected.

Comment 6: “In lines 55-57, the authors should add that this review aims also to provide recommendations for future research in this field.”

Answer 6: The specified objective has been added.

Comment 7: “In lines 71,72, “Table 1. Gender distribution, birth weight, birth length, and birth OFC (cm) in case reports of patients with Desbuquois Syndrome.”, please be consistent and either add reference units such as g and cm where missing or leave out at all.”

Answer 7: Units have been provided at the top of the first table.

Comment 8: “Subsection 3, “Symptom Occurrence by Dysplasia Type”, could be changed to “Clinical characteristics in Desbquois Dysplasia”.”

Answer 8: In accordance with the received advice, the section title has been changed.

Comment 9: “Subsection 4 “Genetic Mutations” could be changed to “Molecular genetics”.”

Answer 9: The title has been changed according to the suggestion, for which we are grateful.

Comment 10: “In Table 2 “Radiolgical features” should be corrected.”

Answer 10: The specified word has been corrected.

 Comment 11: „In Table 2, when summarising the symptoms, did the authors consider if something was not reported because it was not present or could not be assessed? It may be worthwhile to specify it.”

Answer 11: All symptoms reported by the authors in the published case descriptions were documented. This information has been added to the introduction.

Comment 12: „Table 2 has an excessive length; several redundant voices could be trimmed or merged.”

Answer 12: Thank you for another valid remark and suggestion. It was also conveyed by the first reviewer. The symptoms in Table 2 have been arranged from the most common to the least common, and the table has been condensed by removing symptoms observed in fewer than approximately 6-7 patients.

Comment 13: “In Table 2, regarding the description of the symptoms, the authors should specify whether the clinical terms used are HPO terms, and if they are, they should also add the appropriate code. I’d recommend using this standardised vocabulary since it will significantly help reduce redundancy (i.e. thin vermilion border/thin lips/thin upper lips;  Small, narrow or bell-shaped thorax/Barrel-shaped thorax; various clinodactily in Table 2). In my opinion, it is an effort that will pay to support future studies, improving biomedical research because it standardises the way human phenotypes are described, ensuring consistency and accuracy across different clinical databases and research studies. Therefore, it becomes easier to analyse and interpret clinical data, leading to more accurate and early therapeutic intervention diagnoses and a better understanding of genotype/phenotype correlations.”

Answer 13: In Table 2, where possible (if a corresponding term existed in the database), symptom names have been updated to HPO terminology, with the appropriate codes included. Synonymous symptoms have been combined. Various forms of clinodactyly were merged into a single entry.

Comment 14: “In Table 3, it would be better to put for consanguinity Y(yes) or N (No), since it is more immediate.”

Answer 14: Thank you for the valuable advice; the entry has been made accordingly.

Comment 15: “In Table 3, "Ethicity" should be corrected and the relative term should be adjusted (i.e Moroccan not Morocco, Turkish not Turkey, Iranian not Iran…)”

Answer 15: In Table 3, the "Ethnicity" column has been corrected to ensure consistent grammatical usage throughout.

Comment 16: “According to HGVS Nomenclature by Den Dunnen et al. (2016), it is recommended to use three-letter amino acid code abbreviations for protein variant descriptions; furthermore, so doing they will be concordant throughout the table and the text. Moreover, all variants should be described with an accepted reference sequence, i.e. NMXXXX. The authors could specify this in a note at the end of the table.”

Answer 16: The "Amino Acid Change" column has been revised to ensure that three-letter amino acid code abbreviations are used consistently for protein variant descriptions. Information about the reference sequences has also been added.

Comment 17: “I suggest adding a column in Table 3 specifying the mutation’s status, i.e., homozygous or compound heterozygous.”

Answer 17: In the "Nucleotide Change" column, information indicating "hom" for homozygous and "chet" for compound heterozygous has been added in parentheses. An explanation of these abbreviations has been provided below the table.

Comment 18: “The authors should add a section as “Molecular pathogenesis” and add something about the protein coded by XYLT1 and its functional role. Are the two proteins, XYLT1  and CANT1, somehow implicated in the same pathway? Are there any connections? They should also add something about the physiopathological mechanisms. Is there a connection between the two proteins, XYLT1 and CANT1, through their roles in glycosaminoglycan (GAG) biosynthesis and proteoglycan metabolism? This information will contribute to a better understanding of the underlying physiopathological mechanisms, making the review more comprehensive and helpful.”

Answer 18: The aspect highlighted by the reviewer has been more thoroughly described in the introduction, as the initial version of the manuscript included basic information in this area. At this point, the description has been expanded.

Comment 19: „A discussion about differential diagnosis and management would be interesting to add in the light of a more comprehensive review. The text could also be changed to be more extensive.”

 Answer 19: We have added a few sentences regarding treatment.
